


# Divergence of dominant factors on soil microbial communities and functions in forest ecosystems along a climatic gradient

Zhiwei Xu [1], Guirui Yu [2,3,*], Xinyu Zhang [2,3,*], Nianpeng He [2,3], Qiufeng Wang [2,3], Shengzhong Wang [1], Xiaofeng Xu[4], Ruili Wang [5], Ning Zhao [6]

College of Geographical Sciences, Northeast Normal University, Changchun, 130024, China
Key Laboratory of Ecosystem Network Observation and Modeling, Institute of Geographic Sciences and Natural Resources Research, Chinese Academy of Sciences, Beijing 100101,China
College of Resources and Environment, University of Chinese Academy of Sciences, Beijing, 100190, China
Biology Department, San Diego State University, San Diego, CA 92182, USA
College of Forestry, Northwest A&F University, Yangling, Shaanxi Province 712100, China
Cold and Arid Regions Environmental and Engineering Research Institute, Chinese Academy of Sciences, Lanzhou 730000, China

* Corresponding author: Key Laboratory of Ecosystem Network Observation and Modeling, Institute of Geographic Sciences and Natural Resources Research, Chinese Academy of Sciences, Beijing 100101, China. No. 11A, Datun Road, Chaoyang District, Beijing, 100101, China. Tel.: +86-10-64889268; fax: +86 10 64889432.

E-mail: yugr@igsnrr.ac.cn (G.Y.), zhangxy@igsnrr.ac.cn (X.Z.)





**Abstract.** Soil microorganisms play an important role in regulating nutrient cycling in terrestrial
ecosystems. Most of the studies conducted thus far have been confined to a single forest biome or
have focused on one or two controlling factors, and few have dealt with the integrated effects of
climate, vegetation, and soil substrate availability on soil microbial communities and functions
among different forests. In this study, we used phospholipid-derived fatty acid (PLFA) analysis to
investigate soil microbial community structure, and extracellular enzymatic activities to evaluate
the functional potential of soil microbes of different types of forests in three different climatic
zones along the North-South transect in eastern China (NSTEC). In general, soil enzyme activities
and microbial PLFAs were higher in primary forests than in secondary forests in temperate and
warm temperate regions. In the subtropical region, soil enzyme activities were lower in the
primary forests than in the secondary forests and microbial PLFAs did not differ significantly
between primary and secondary forests. The microbial PLFAs and enzyme activities differed
considerably between broadleaved and coniferous forests. Different species of coniferous trees
may cause variations in soil microbial PLFAs and enzyme activities. Both climate and forest type
had significant effects on soil enzyme activities and microbial communities with a considerable
interactive effect. Litter nutrients made an important contribution to variations in the soil
microbial communities and enzyme activities in temperate zones, while soil micro-climate and
nutrients were the main controls on the soil microbial community structure and enzymatic
activities in warm temperate and subtropical zones. Our results indicate that the main controls on
soil microbes and functions vary across forest ecosystems in different climatic zones, and that the
effects of soil moisture content, soil temperature, and the soil N/P ratio were considerable. This
information will add value to modeling of microbial processes and will contribute to carbon
cycling in large-scale carbon models.



## 1 Introduction


There is a growing awareness that above- and below-ground interactions make an essential
contribution to ecosystem function (van Dam and Heil, 2011). Variations in soil microbial
diversity and community structure have a strong influence on soil organic matter turnover and
may impact on the function of a given ecosystem (Baumann et al., 2013). For example,
mycorrhizal fungi and nitrogen (N) fixing bacteria are responsible for 80% of all N, and up to 75%
of phosphorus (P), that is acquired by plants annually (van der Heijden et al., 2008). Therefore, it
is important to study the composition and enzyme activities of soil microbial communities to
obtain an improved understanding of the mechanisms that control soil organic carbon dynamics in
different forest ecosystems.
Vegetation composition may alter soil physicochemical properties by changing the quantity
and quality of plant litter, which further influence microbial community composition and function
(Ushio et al., 2010; Deng et al., 2015). There is increasing evidence that vegetation types influence
the structure and functions of the soil microbial community (Yin et al., 2014; Zheng et al., 2015).
Differences in microbial communities, as represented by PLFAs, have also been reported among
adjacent maple, beech, hornbeam, lime, and ash forests in Germany (Scheibe et al., 2015) and
among forests of four conifer species in coastal British Columbia (Grayston and Prescott, 2005).
From a functional perspective, both soil acid phosphatase and b-glucosidase activities were higher
in a monsoon evergreen broadleaf forest than in a Masson pine forest (Zheng et al., 2015).
However, vegetation type does not always have an effect on the composition of the soil microbial
community. Hannam et al. (2006) reported that the microbial community composition of a white
spruce-dominated forest differed substantially from that of an aspen-dominated stand, but was
similar to that of a mixed stand with equivalent proportions of deciduous and coniferous trees.
Most of the studies conducted thus far have been confined to a single forest biome or have focused
on one or two controlling factors (Ultra et al., 2013), and few have dealt with the integrated effects
of climate, vegetation, and soil substrate availability on soil microbial communities and functions
in different forest biomes.
Soil microbial communities and enzyme activities can be influenced by an array of factors,
such as climate (Xu et al., 2015), vegetation types (Urbanováet al., 2015), plant diversity (Li et al.,





2015), and physico-chemical soil properties (Tripathi et al., 2015). The links between the diversity
of plant and soil microbial communities and enzyme activities are widely acknowledged (Chung et
al., 2007). The composition of the vegetation species can be used to successfully predict the soil
microbial community (Mitchell et al., 2010). Soils with different vegetation types develop distinct
physico-chemical properties that will have pronounced effects on the structure and function of the
soil microbial community (Priha and Smolander, 1997). Soil organic matter is related to the
variations in microbial activities and community function (Brockett et al., 2012). Soil pH (Shen et
al., 2013), elemental stoichiometric ratios (Högberg et al., 2007), and nutrient status (Lauber et al.,
2008) have also been identified as determinants of microbial community structure. However, we
still do not know which mechanisms control the variability in the structure and functions of soil
microbial communities within different groups of plant species (broadleaved and coniferous trees)
on similar soil types within the same climatic region.

Forest soil microbial community structures and enzyme activities are influenced by different

factors in different climatic zones. For example, Högberg et al. (2007) found that the soil
microbial community composition in a boreal forest was strongly influenced by the soil carbon to
nitrogen ratio (C/N) and the soil pH. Studies in temperate forests have shown that dehydrogenase
and urease were closely related to the mean air temperature, litter production, and nutrient
availability (Kang et al., 2009). In addition, Hackl et al. (2005) reported that soil water availability
was responsible for variability in the microbial community structure of temperate forests.
Precipitation and soil moisture may be important controls on the structure of soil fungal
communities of tropical forests (Eaton et al., 2011; McGuire et al., 2012). However, there is a lack
of well-defined information about the factors that influence the structure and functions of soil
microbial communities in forests with different plant species (broadleaved and coniferous trees)
across a range of climates and soils.

The North-South Transect of Eastern China (NSTEC) represents a latitudinal and climatic

gradient. It is a unique belt in which vegetation ranges from boreal forest to tropical rain forest,
depending on the local temperature and precipitation conditions. In this study we examined
variations in the soil microbial communities and their functions in forests comprising different
species (broadleaved and coniferous trees) in temperate, warm temperate, and tropical forest



biomes along the NSTEC. The temperature and precipitation are different in these three climatic
zones. We used information about the soil physico-chemical properties, microbial community
structure, and hydrolytic enzyme activities involved in C, N, and P transformations to explore how
soil microbial communities and enzyme activities differed among different forest types in different
climatic zones, and to determine the influence of different environmental variables on the soil
microbial communities and enzyme activities in different climatic zones.
**2 Materials and methods**
**2.1 Study area and soil sampling**
We chose three study sites, namely Liangshui in Northeast China, Taiyue Mountain in North
China, and Dinghu Mountain in South China, along the North-South Transect in Eastern China
(NSTEC) for field measurements and soil sampling (Fig. 1). Both the air temperature and
precipitation decrease from south to north along the NSTEC (Table 1).
We examined all the representative forest species in each climatic zone. In Liangshui, on the
Xiao Xing'an Mountain, we sampled primary conifer broad-leaved mixed forest (PCB), secondary
conifer broad-leaved mixed forest (SCB), and two coniferous plantations, one of which was
mainly *Pinus koraiensis* (PK) while the other was *Larix olgensis* (LO). On Taiyue Mountain, we
sampled primary deciduous broad-leaved forest (PDB), secondary deciduous broad-leaved forest
(SDB), and two coniferous plantations, one of which was comprised mainly of *Pinus*
*tabulaeformis* (PT) while the other was mainly *Larix olgensis* (LO). On Dinghu Mountain, we
sampled a primary evergreen broadleaved forest (*Castanopsis chinensis*, *Cryptocarya chinensis*,
*Cryptocarya concinna*, *Erythrophleum fordii*, and *Cyathea podophylla*), secondary conifer and
broadleaf mixed forest (*Pinus massoniana, Schima superba*), aconiferous plantation (*Pinus*
*massoniana*), and an evergreen broadleaved plantation (*Erythrophleum fordii*) along a
successional stage, hereafter referred to as PEB, SCB, PM, and EF, respectively. The primary
forests are zonal forests that reflect the regional climate and the others are zonal forests that reflect
the extreme site conditions. Information about the climate, soil classification (Soil Survey Staff
2010), and soil properties at each site is provided in Table 1.
Soil samples were collected at nine sampling sites along the NSTEC in July and August 2013.
Each site had four independent plots in well-drained areas, which covered an area of 30 m × 40 m,





and were at least 10 m apart. The vegetation composition of the four plots at each site was similar.
Samples of mineral soil were collected from a depth of 0–10 cm at between 30 and 50 points in
each plot along an S-shape using a custom-made coring device with a diameter of 6 cm. The
above-ground standing biomass, dead plant parts, and litter were removed from each sampling
point. These samples were pooled together as a composite sample. Visible roots and residues were
removed and then the soil fractions of each sample were homogenized.

We stored the samples at 4 ℃ in a portable refrigerator during field sampling. Once returned

to the laboratory, samples were stored at 4 ℃ before analysis. Soils were analyzed for enzyme
activities and PLFAs in September 2013. The fresh soil samples were sieved through a 2-mm
mesh and were subdivided into three subsamples. One subsample was stored at 4 ℃ until
analyzed for soil enzyme activities and physical and chemical properties. The second was stored at
−20 ℃ before analysis for microbial community structures. The third was air dried, and then
sieved through a 0.25 mm mesh before SOC, TN, and TP analysis. The soil temperatures were
measured *in situ* at the time of sampling. Soil moisture content (SMC) was measured
gravimetrically on 20 g fresh soil that was oven-dried at 105 ℃ to constant weight immediately on
arrival at the laboratories at the study sites (Liu et al., 2012).
**2.2 Soil chemical analyses**
Soil pH was measured at a soil-to-water ratio of 1:2.5. Soil total N (TN) concentrations were
determined by dry combustion of ground samples (100-mesh) in a C/N analyzer (Elementar, Vario
Max CN, Germany). The soil organic carbon (SOC) concentrations were determined by
dichromate oxidation and titration with ferrous ammonium sulfate (Huang et al., 2014). The litter
total C (litter TC) and total N (litter TN) were determined with the same method that was used for
soil TN. Total phosphorus (TP) was determined with a flow injection auto-analyzer following
digestion with $H_2SO_4$-$HClO_4$ (Huang et al., 2011). The soil clay fraction (hereafter referred to as
Clay, comprised of particles <53 μm) was separated by wet-sieving and then freeze-dried (Six,
Elliott & Paustian 2000).
**2.3 Phospholipid fatty-acid and enzyme activity analysis**
Samples were analyzed for phospholipid fatty-acids (PLFA) using the method described by B ååth
& Anderson (2003). After mild alkaline methanolysis to form fatty acid methyl esters (FAMEs),



samples were then dissolved in hexane and analyzed with a DB-5 column in a gas chromatography
mass spectroscopy (GCMS) system (Thermo TRACE GC Ultra ISQ). Total amounts of the
different PLFA biomarkers were used to represent the different groups of soil micro-organisms
(Table S1). Taken together, the combination of bacterial, fungal and actinomycic PLFAs
biomarkers represented the total PLFAs of the soil microbial community.

The activities of β-glucosidase (BG), N-acetylglucosaminidase (NAG), acid phosphatase

(AP), and leucine aminopeptidase (LAP) were measured as outlined by Saiya-Cork, Sinsabaugh &
Zak (2002). The microplates were incubated in the dark at 20 ℃ for 4 h. During the incubation,
the incubation plates were shaken every hour to ensure the reaction mixtures were homogenous.
Fluorescence was measured using a microplate fluorometer with 365-nm excitation and 450-nm
emission filters (Synergy[H4] Hybrid Reader, Synergy[H4] BioTek, USA).
**2.4 Statistical analysis**
One-way analysis of variance (ANOVA) with a post-hoc Tukey HSD test was used to test the
differences between the soil and microbial properties in the various forests of the three climatic
zones. All data were normality distributed. Two-way analysis was used to test the effect of climate
and vegetation on the soil microbial properties. All ANOVA and two-way analysis were
performed using SPSS 19.0 for Windows. Figures were generated using the Origin 8.0 package.
Data are reported as the mean $\pm$ SE.

Redundancy analysis (RDA) was used to examine the relationships between the litter factors

(litter TC, litter TN, litter C/N), soil biochemical variables (soil temperature (ST), soil moisture
content (SMC), pH, C/N, soil carbon to phosphorus ratio (C/P), soil nitrogen to phosphorus ratio
(N/P), SOC, TN, TP), soil texture (Clay), and the soil microbial community compositions and
enzyme activities. Before redundancy analysis, we conducted forward selection of the
environmental variables that were significantly correlated with variations in the microbial
communities and enzyme activities using stepwise regression and the Monte Carlo Permutation
Test that was similar to the multiple regression analysis. Stepwise regression and RDA were
processed using CANOCO software 4.5 (Ter Braak & Smilauer 2002). The vectors of greater
magnitude that formed smaller angles with an axis were more strongly correlated with that axis.
**3 Results**



### 3.1 Soil enzyme activities in different vegetation types


The soil enzyme activities were generally higher in the primary forests than in the secondary
forests in temperate and warm temperate climatic zones (Fig. 2). However, in the subtropical
climatic zone, soil enzyme activities were higher in the SCB forest than in the PEB forest. The BG,
NAG, and AP enzymes in the two soils of the PT and LO in the warm temperate zone were
significantly different (Fig. 2(A, B, D)). Climate, a significant influence on the variations of soil
enzyme activities ($P<0.0001$), had more influence than forest type. The effects of climate and
forest type interactions were only significant for soil NAG ($P<0.0001$) and AP activities ($P=0.035$)
(Table 2, Table S2).

### 3.2 Soil microbial community composition in different vegetation types


Soil PLFAs were higher in the primary forest in the temperate and warm temperate zones than in
the secondary forest. In the temperate zones, soil PLFAs were higher in the PCB forest than in the
SCB, PK, and LO (Fig. 3A). In the warm temperate forests, total soil microbial PLFAs were
highest in the LO forest (Fig. 3B). In the subtropical zone, total, bacterial, and actinomycic PLFAs
were higher in the PEB and SCB forests than in the PM and EF forests (Fig. 3C).
Climate had a significant effect on the Total PLFAs, fungi, and G$^-$ ($P<0.0001$), and the forest
type had a significant effect on the soil bacteria, fungi, G$^+$, and G$^-$ PLFAs. With the exception of
the soil G$^+$/ G$^-$, the effect of the combination of climate + forest type on all soil PLFAs was
significant, and was stronger than the individual effects of either climate or forest type (Table 2,
Table S2). The soil microbial communities in the different forests in the three climate zones were
generally unique (Fig. 5).

### 3.3 Relationships between soil enzyme activities and soil properties


The litter C/N, litter TN, and SMC ($P=0.002$) were the most important influences on the soil
enzyme activity variations in the temperate forests, followed by ST, soil N/P, and soil TP (Fig.
4(A)). In the warm temperate forests, the variations in the soil enzyme activities were significantly
and positively correlated with ST and soil pH ($P=0.002$), but were negatively correlated with SMC
and soil nutrients (TN and SOC) (Fig. 4(B)). In the subtropical forests, soil enzyme activities were
significantly and positively correlated with clay, SMC, soil TN, and TP ($P=0.002$), followed by
soil nutrient ratios (Fig. 4(C)). These results indicate that the litter inputs, soil micro-climate, and





soil texture were the main drivers of variations in the soil enzyme activities in the temperate, warm
temperate, and subtropics, respectively, with ST, pH, SMC, and soil N/P as additional influences.

**3.4 Relationships between plfa profiles and measured soil properties**

In the temperate forests, the variations in the soil microbial community structure were strongly
affected by the litter TN, litter TC, litter C/N, soil TP, and ST ($P$=0.002) (Fig. 5(A)). In the warm
temperate forests, the first axis of the RDA plot of the soil microbial community structure was
significantly and positively correlated with ST ($P$=0.002), but was negatively correlated with soil
N/P, soil TN, soil C/P, and SOC ($P$=0.002) (Fig. 5(B)). In subtropical forests, the variations in the
soil microbial community structure were significantly and positively correlated with litter TC and
ST ($P$=0.002), but negatively correlated with SMC, soil C/P, soil N/P, and soil C/N ($P$=0.002),
followed by the soil TN and clay contents (Fig. 5(C)). The litter C/N was the main influences on
the variations in the soil microbial communities in the temperate, and the soil N/P was the main
influences in the warm temperate and subtropical forests. The microbial communities were also
influenced by ST, pH, SMC.

**4 Discussion**

**4.1 Response of soil enzyme activities and microbial plfas to variations in forest type**

As expected, soil enzyme activities differed between the coniferous, deciduous, and broad-leaved
forests in the three climatic zones. The PCB in the temperate zone is a conifer broad-leaved mixed
forest and has higher inputs of mixed litter than a single species coniferous forest (Zhang et al.,
2008). Therefore, all enzyme activities were highest in PCB in the temperate zone. The higher soil
enzyme activities in the coniferous forests relative to those in the deciduous broad-leaved forests
in the warm temperate zone reflect the high SOC and TN concentrations in the two coniferous
forests (Table 1). Extracellular enzymes catalyze the rate-limited steps of decomposition and
nutrient cycling (Koch et al., 2007), thereby improving the soil nutrient availability. The soil
enzyme activities were highest in the SCB forest, reflecting the higher soil nutrient concentrations
in subtropical zones.
The soil microbial community structures under the various forest types differed significantly
across the three climatic zones. Vegetation transfers substrate material of varying quality to
microbes through litter fall. Litter from broadleaved forests typically contains high levels of




water-soluble sugar, organic acid, and amino acids (Priha and Smolander, 1997; Priha et al., 2001),
and promotes the propagation of bacteria that favor high-nutrient soil. However, fungi are mainly
responsible for lignin degradation and are presumably more capable of coping with the
degradation of pine litter that contains high amounts of recalcitrant polymeric phenolic
compounds such as lignin and tannin than bacteria (Wardle et al., 2003; Hackl et al., 2005).
Therefore, the structures and functions of the soil microbial communities that developed in the
different types of forest were unique.
The variations in the plant functional traits between the different forest types, especially
between the deciduous and coniferous forests, will promote the development of different soil
microbial communities. Several other studies have described how SLA, LDMC, and leaf N
influence soil microbial community structure and function (Orwin et al, 2010; de Vries et al., 2012;
Pei et al., 2016). While plant trait data were not available for this study, our results were similar to
those from other studies of the nine primary forests along the NSTEC (data have not been
published). Despite this, climatic region may have more influence on soil enzyme activities and
soil microbial communities than forest type, and other studies have reported how climate
influences the large-scale distribution of microorganisms (de Vries et al., 2012; Xu et al., 2017).
**4.2 Common influences on soil enzyme activities and microbial communities**
Although soil microbial communities and functions varied between the different forests, they were
subject to some common influences. For example, our results showed that ST, SMC, soil pH, and
soil N/P ratio influenced, but perhaps did not dominate, the responses of the soil microbial
community structures and enzyme activities in the different forest types across the three climatic
zones.
Temperature can influence enzyme activity directly and indirectly by modifying the enzyme
kinetics and influencing the proliferation of microbes, respectively (Kang et al., 2009). By
changing the quality and quantity of the substrate on which microbes function, soil moisture is an
important driver of the overall microbial composition and soil microbial function (Hackl et al.,
2005). The responses of soil enzyme activities and microbial communities in the various forest
types were all significantly influenced by the SMC in the three climatic zones. Increases in soil
moisture can enhance both the release and the diffusion rates of enzymes, substrates, and reaction



products (Burns et al., 2013), and our results showed that soil enzyme activities and microbial
PLFAs increased as the SMC increased in the warm temperate and subtropical zones. However,
water-logged conditions are not suitable for microbes and are not beneficial for the release of soil
enzymes (Lucas-Borja et al., 2012), and, similar to other studies, soil enzyme activities and SMC
were negatively correlated in the temperate zone forests (Brockett et al., 2012). As the SMC
increases, the bacterial PLFAs increase (Myers et al., 2001) and fungal PLFAs decrease (Staddon
et al., 1998), which indicates that the soil microbial communities and enzyme activities in the
different climatic zones were all influenced by the soil micro-climate. This was also demonstrated
by the stronger effect of climate on soil enzyme activities and the combined interaction effect of
climate and forest type on soil microbial communities.
An increasing number of studies has reported that the soil microbial composition and enzyme
activities are largely related to soil pH at continental (Fierer and Jackson, 2006) and global scales
(Sinsbaugh et al., 2008). Soil pH directly affects the activities of extracellular enzymes
immobilized in the soil matrix, and the effect of soil pH on the soil microbial community and
function reflects the influence of vegetation through changes in soil chemistry. Every enzyme has
a well-defined optimal soil pH value (Sinsabaugh et al., 2008) that results from different levels of
soil enzyme activities under different soil pH conditions. Increases in pH lead to increases in
bacterial diversity and cause the bacterial community to shift, so that there are more $G^-$, and less
$G^+$, bacteria PLFAs (Wu et al., 2009; Shen et al., 2013).
Many other studies have reported how different factors determine the response of the soil
microbial community and function to variations in forests (Högberg et al., 2007; Kang et al., 2009;
Eaton et al., 2011; McGuire et al., 2012). Mostly limited to one climatic zone, these studies were
quite diverse and featured a range of microbial methods, sampling times, and environmental
properties, which means it is difficult to compare the results. In this study, we collected the
samples at the same times and used the same methods to analyze the soil microbial communities
and enzyme activities. We found that the different climatic zones shared common factors that
influenced the responses of the soil microbial communities and functions to forest variations. Soil
microbes have unique roles in C, N, and P cycling that depend on the vegetation type and soil
properties (Sugihara et al., 2015; Wu et al., 2015). Our results suggest that the nutrient cycling



mechanisms probably vary between different vegetation types and climatic zones; however,
further studies are needed to define the patterns and drivers of nutrient cycling.
**4.3 Key influences on soil enzyme activities and microbial communities**
Our results showed that the most important controls on the responses of soil microbial
communities and enzyme activities to vegetation types varied across climatic zones. The litter
quality and quantity contribute to the maintenance of soil fertility in forest ecosystems (Wang et al.,
2011). In our study, and the C/N ratios were highest, in litter from PCB stands (Table 1), which
shows that the soil in the PCB was more N-limited than the other soils because of litter inputs with
high C/N ratios (Table 1). Therefore, the microbial N demand was highest in soil in the PCB forest,
which resulted in higher NAG and LAP values. Plant litter has a strong influence on soil microbial
composition and activity, as the litter decomposition process provides nutrients for microorganism
growth through inputs of leaf litter (Attiwill and Adams, 1993), dying roots (Silver and Miya,
2001), and root secretion (Grayston et al., 1997). The litter from the mixed forests, represented in
our study by PCB, is more diverse than that from the pure forests, and so a wider variety of soil
microbes participate in the decomposition process, so that the soil organic matter is richer, and
there are more soil microbial PLFAs, than in the other forest types. Fungi typically dominate
N-limited environments and the fungal biomass is positively related to the C/N ratio (Nilsson et al.,
2012). The F/B ratio was therefore highest in the PCB forest where the litter C/N values were
highest.
The soil N/P ratio was the most important influence on the soil microbial communities and
enzyme activities in the warm temperate zone, which is consistent with the results of previous
studies (Shen et al., 2013; Högberg et al., 2007). Soil stoichiometric C, N, and P ratios reflect the
nutrient limitations of the ecosystems (Sterner and Elser, 2002) and should indicate soil organic
matter mineralization and sequestration (Gundersen et al., 1998). Soil microorganisms obtain C, N,
and P in such a way that enzyme release corresponds with the soil stoichiometric ratios of C, N,
and P. When supplies of N or P are limited, the activities of the enzymes that are responsible for
nitrate or phosphate mineralization will be higher. Consistent with this discussion, soil enzyme
activities in subtropical forests (DH) responded positively to the soil C/N and N/P ratios.
Microbes obtain the nutrients they need to construct biomass by decomposing soil organic



matter. Wallenius et al. (2011) found that the soil bacterial biomass was higher in forests where the
soil organic matter concentrations were higher than in forests with low soil organic matter
concentrations, and Xu et al. (2017) found positive relationships between soil enzyme activities
and SOC and TN concentrations along the NSTEC. In line with the resource limitation model, and
also confirmed by several other studies (Brockett et al., 2012; Zhang et al., 2013), Schimel and
Weintraub (2003) suggested that increases in N and C substrate availability might favor enzyme
synthesis. Soil microorganisms however did not grow when the available P concentrations in soil
were less than 0.7 mg kg$^{-1}$ and were stimulated by P additions (Zheng et al., 2009). Other studies
have reported that P additions stimulated the different PLFA microbial groups in soils (Dong et al.,
2015). The positive correlations between both total microbial biomass and microbial composition
and available P suggests that microbes may be dependent on the P supply in some forest
ecosystems, especially in subtropical forests (DeForest et al., 2012; Zhang et al., 2013; Xu et al.,

2017).

The soil clay content had most influence on the soil enzyme activities in subtropical forests.

Soil texture is a key property that affects the accessibility of organic matter to microbes, and is an
important determinant of soil moisture, and nutrient availability and retention (Veen and Kuikman,
1990). Consistent with our results, Lagomarsinoa et al. (2012) reported that the activities of soil
BG, AP, and NAG were higher in silt and clay fractions than in coarser fractions. This may be
attributed to the presence of clay-humus-enzyme complexes in the finest soil fractions, and
implies that physical protection affects soil enzyme activities. In addition, fine textured soils with
higher silt and clay contents are known to be more conducive to bacterial growth than coarser soils
because they have a greater water-holding capacity, higher nutrient availability, and offer better
protection against bacterial grazers (Carson et al., 2010).
**4.4 Implications for ecosystem modeling**
There is increasing recognition that, to improve climate models, microbial processes should be
simulated (DeLong et al., 2011; Xu et al., 2014). As such, this study has three important
implications. First, microbial datasets that have information about enzyme activities and soil
microbial properties contribute to improved parameterization of ecosystem models (Xu et al.,
2013; 2017). Information about the spatial patterns of, and factors that control, microbial





properties and enzymatic activities can enrich the datasets that are used to parameterize models of
microbial processes (Wang et al., 2013; Allison et al., 2010). Secondly, knowledge about
microbial community structure and its environmental controls can give a better understanding of
how microbes adapt to changing environments, which is the main direction of model development
(Schimel and Schaeffer, 2012). Information about edaphic controls on microbial processes is
critical for developing new modeling frameworks with improved links with field experimental
data (Abramoff et al., 2017). Finally, the information generated in this study about the divergence
of the dominant factors that control soil microbial properties across forests is extremely valuable
for improving our understanding of soil microbial ecology and forest management.
**5 Conclusions**
In this study, we characterized the soil microbial communities and enzyme activities and factors
that controlled them in various forest types across three different climatic zones. We found that
forest types with specific soil conditions supported the development of distinct soil microbial
communities with variable functions. The litter TN, soil temperature, and soil clay contents were
important predictors of the variance in soil enzyme activities in temperate, warm temperate, and
subtropical zones, respectively, while litter and soil nutrient ratios were significant predictors of
the variance in soil microbial communities. We also found that SMC, soil temperature, soil pH,
and the soil N/P ratio were common drivers of variations in the soil microbial community structure
and enzyme activities across the different forest types in the three climatic zones. The data in this
study is extremely valuable for improving our understanding of soil microbial ecology and forest
management.
*Data accessibility*. Requests for data and materials should be addressed to N.H. (henp@igsnrr.ac.cn) and G.Y.
(yugr@igsnrr.ac.cn).

*Author contributions*. Z.W.X., G.R.Y. and X.Y.Z. planned and designed the research. Z.W.X., N.P.H., R.L.W.,
N.Z., C.C.J., and C.Y.W. conducted fieldwork. Z.W.X., G.R.Y., X.Y.Z. Q.F.W., S.Z.W. and X.F.X wrote the
manuscript. All authors contributed critically to the drafts and gave final approval for publication.





*Competing interests.* The authors declare that they have no conflict of interest.

*Acknowledgements.* We thank Dr. Wenyi Dong for assisting with phospholipid fatty acid analysis, and Ms Jinfeng
Bu for assisting with soil enzyme activity analysis. This study was conducted at the three field stations along the
North-South Transect in Eastern China (NSTEC), and we thank the field station staff for their assistance with
sampling and measurements. This research was jointly supported by the Key Program of the National Natural
Science Foundation of China (31290221, 31290222), the National Natural Science Foundation of China
(41601084), and the Fundamental Research Funds for the Central Universities (2412016KJ029). X.X. was grateful
for the financial support from the San Diego State University and the Oak Ridge National Laboratory.

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





## Figure captions

**Figure 1.** Distribution of typical forest ecosystems along the North-South Transect of Eastern China (NSTEC). The names of the sampling sites from north to south were abbreviated as followed: LS = Liangshui; TY = Taiyue; DH = Dinghu.

**Figure 2.** Soil enzyme activities under different forest types in different climatic zones (A. Liangshui; B. Taiyue; C. Dinghu). Different lowercase letters indicate significant differences between forests in the same climatic zone. The abbreviations of the sampling sites are shown in Table 1.

**Figure 3.** The PLFA contents, Fungi:Bacteria ratios, and $G^+/G^-$ for different forest types in different climatic zones (A. Liangshui; B. Taiyue; C. Dinghu). Different lowercase letters indicate significant differences among forests in the same climatic zone. F/B, fungi/bacteria; $G^+/G^-$, Gram-positive bacteria/ Gram-negative bacteria. The abbreviations of the sampling sites are shown in Table 1.

**Figure 4.** Redundancy analysis (RDA) ordination biplot of soil enzyme activities and environmental properties for the different forest types in different climatic zones (A. Liangshui; B. Taiyue; C. Dinghu). Only the environmental variables that were significantly correlated with RDA1 are shown. The dotted lines and solid lines represent the environmental variables and enzyme activities. The variables in this table were abbreviated as follows: TC(litter) = litter total carbon; TN(litter) = litter total nitrogen; C/N(litter) = litter total carbon/nitrogen; ST = soil temperature; SMC = soil moisture content; Clay = soil clay content; SOC = soil organic carbon; TN = soil total nitrogen; TP = soil total phosphorus; C/N = soil carbon/nitrogen; C/P = soil carbon/phosphorus, and N/P = soil nitrogen/phosphorus.

**Figure 5.** Redundancy analysis (RDA) ordination biplot of soil microbial community structure and environmental properties for different forest types in different climatic zones (A. Liangshui; B. Taiyue; C. Dinghu). Only the environmental variables that were significantly correlated with RDA1 are shown. The dotted lines and solid lines represent the environmental variables and lipid signatures. The abbreviations of the variables included in this figure are shown in Figure 4.

## Supporting Information

**Table S1.** The PLFA biomarkers used to represent the different groups of soil micro-organisms (Frostegård *et al*.1996).

**Table S2.** Average values of soil enzyme activities and microbial PLFAs in the three different climatic zones and three different forest types, respectively.



**Tables**


**Table 1.** Stand characteristics and soil properties under different forest types in the three climatic zones

| Areas[a] | XiaoXing'an Mountain (LS) | | | | Taiyue Mountain (TY) | | | | Dinghu Mountain (DH) | | | |
|---|---|---|---|---|---|---|---|---|---|---|---|---|
| Latitude (°) | 47.19 | | | | 36.70 | | | | 23.17 | | | |
| Longitude (°) | 128.90 | | | | 112.08 | | | | 112.54 | | | |
| Climatic zone | Temperate | | | | Warm temperate | | | | Subtropical | | | |
| MAT (°C) | 0.3 | | | | 6.2 | | | | 20.9 | | | |
| MAP (mm) | 676 | | | | 662 | | | | 1927 | | | |
| Altitude (m) | 401 | | | | 1668 | | | | 240 | | | |
| Soil type | Cryumbrept | | | | Eutrochrepts | | | | oxisol | | | |
| Vegetation type[b] | PCB | SCB | PK | LO | PDB | SDB | PT | LO | PEB | SCB | PM | EF |
| pH | 6.17a | 5.68b | 6.01a | 6.28a | 6.85c | 7.70a | 7.20b | 6.78c | 5.43a | 5.38a | 5.21b | 5.07b |
| ST (°C) | 15.87a | 15.11b | 15.33b | 16.13a | 16.00b | 24.04a | 16.37a | 15.33b | 24.40b | 24.59b | 25.34a | 25.39a |
| SMC (%) | 46.94c | 69.97a | 50.7b | 57.95c | 36.01a | 22.66c | 27.89b | 34.87a | 37.84b | 44.76a | 26.67b | 30.20b |
| Clay (%) | 63.98a | 55.92b | 64.57a | 64.30a | 49.39a | 52.13a | 35.69b | 53.90a | 49.74b | 76.05a | 45.05d | 52.31c |
| SOC (g kg⁻¹) | 62.08a | 75.23a | 61.47a | 57.10a | 41.34a | 17.87b | 42.72a | 42.15a | 28.47b | 40.03a | 26.83c | 37.99b |
| TN (g kg⁻¹) | 4.59a | 4.57a | 4.01a | 4.54a | 2.43b | 1.41c | 3.09a | 2.79a | 1.77b | 2.55a | 1.26c | 1.83b |
| TP (g kg⁻¹) | 0.59b | 0.78a | 0.83a | 0.94a | 0.52b | 0.51b | 0.56a | 0.52b | 0.20c | 0.26a | 0.23b | 0.22b |
| Litter C/N | 43.11a | 24.03c | 31.96b | 25.54c | 48.56b | 37.82c | 53.16a | 30.82d | 28.67a | 27.06a | 30.31a | 29.85a |

[a] PCB, SCB, PK, and LO represent primary conifer broad-leaved mixed forest, secondary conifer broad-leaved mixed forest, *Korean pine* forest and *Larix olgensis* forest, respectively. PDB, SDB, PT, and LO represent primary deciduous broad-leaved forest, secondary deciduous broad-leaved forest, *Pinus tabulaeformis* forest and *Larix olgensis* forest, respectively. PEB, SCB, PM, and EF represent primary evergreen broadleaved forest, secondary conifer and broadleaf mixed forest, *Pinus massoniana* forest and *Erythrophleum fordii* forest, respectively. MAT and MAP indicate mean annual air temperature and mean annual precipitation, respectively; ST, soil temperature; SMC, soil moisture content; SOC, soil organic carbon; TN, soil total nitrogen; TP, soil total phosphorus; Clay, soil clay content; litter C/N, total carbon/total nitrogen of litter.






**Table 2.** The effect of forest types and climate on the soil enzyme activities and PLFAs

| Treatment | | Climate | | Forest type | | Climate × Forest type | |
|---|---|---|---|---|---|---|---|
| | | F | P | F | P | F | P |
| Enzyme activity | BG | **30.487** | **<0.0001** | **6.852** | **0.003** | 3.105 | 0.056 |
| | NAG | **32.793** | **<0.0001** | 5.183 | 0.10 | 3.635 | **0.035** |
| | LAP | **171.864** | **<0.0001** | **16.364** | **<0.0001** | 1.813 | 0.176 |
| | AP | **95.070** | **<0.0001** | **48.117** | **<0.0001** | **22.446** | **<0.0001** |
| PLFAs | tPLFA | **7.764** | **0.001** | 2.697 | 0.079 | **8.666** | **0.001** |
| | Bacteria | 2.796 | 0.073 | **4.921** | **0.012** | **8.357** | **0.001** |
| | Fungi | **8.002** | **0.001** | **21.255** | **<0.0001** | **25.023** | **<0.0001** |
| | Actinomycetes | 0.533 | 0.591 | 2.979 | 0.062 | **3.500** | **0.040** |
| | F/B | **3.731** | **0.032** | **15.502** | **<0.0001** | **6.378** | **0.004** |
| | G⁺ | 0.603 | 0.552 | **3.395** | **0.043** | **5.934** | **0.005** |
| | G⁻ | **12.503** | **<0.0001** | **6.890** | **0.003** | **11.106** | **<0.0001** |
| | G⁺/ G⁻ | 1.662 | 0.202 | 0.069 | 0.933 | 2.257 | 0.117 |

The abbreviations of the variables included in this table are shown in Figure 2 and 3.









**Figure 1.** Distribution of typical forest ecosystems along the North-South Transect of Eastern China (NSTEC). The names of the sampling sites from north to south were abbreviated as followed:

LS = Liangshui; TY = Taiyue; DH = Dinghu.







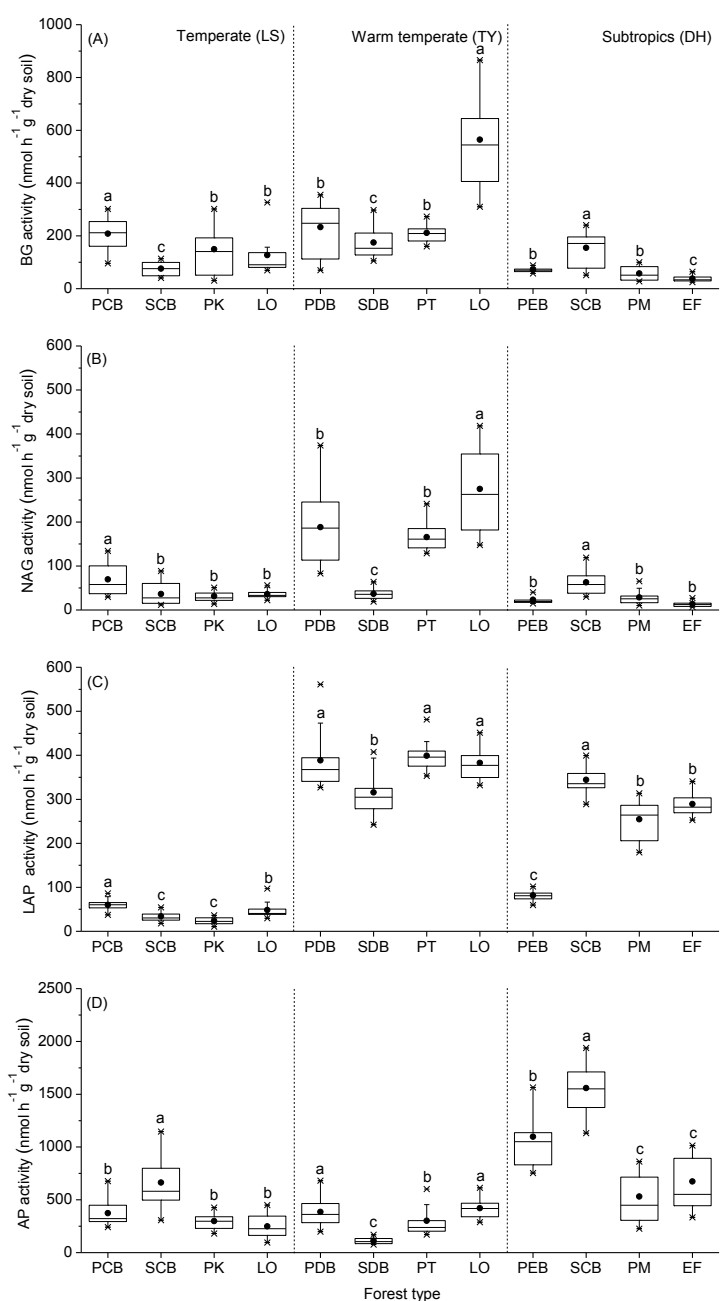


**Figure 2.** Soil enzyme activities under different forest types in different climatic zones (A. Liangshui; B. Taiyue; C.
Dinghu). Different lowercase letters indicate significant differences between forests in the same climatic zone. The
abbreviations of the sampling sites are shown in Table 1.





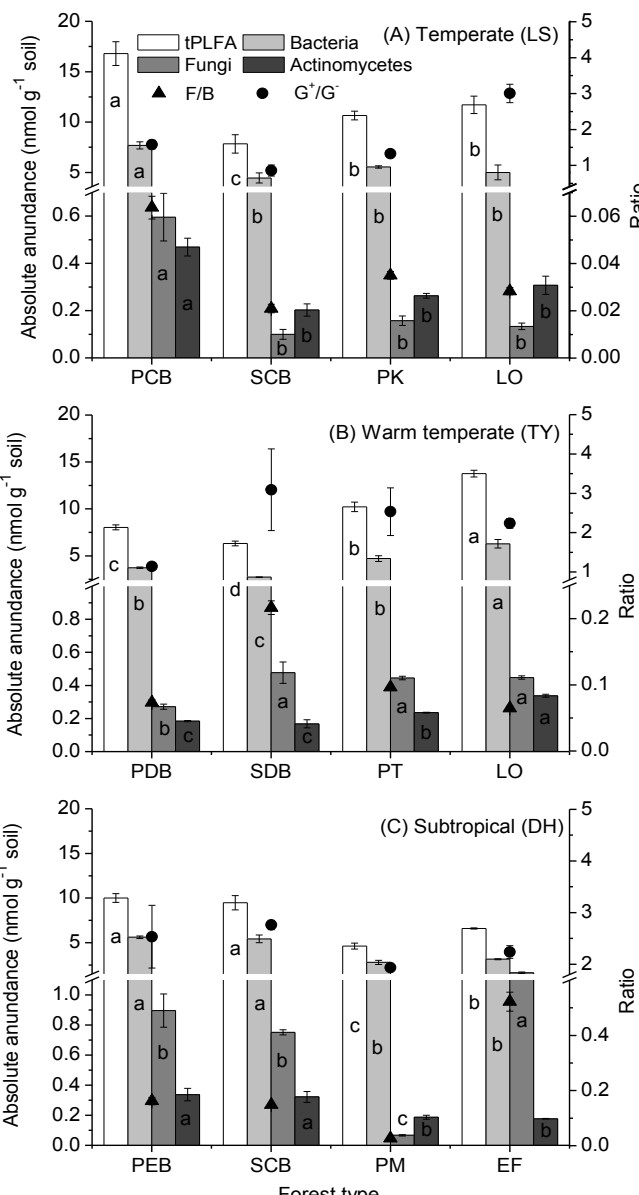


**Figure 3.** The PLFA contents, Fungi:Bacteria ratios, and G+/G− for different forest types in different climatic zones

(A. Liangshui; B. Taiyue; C. Dinghu). Different lowercase letters indicate significant differences among forests in

the same climatic zone. F/B, fungi/bacteria; G+/G−, Gram-positive bacteria/ Gram-negative bacteria. The

abbreviations of the sampling sites are shown in Table 1.



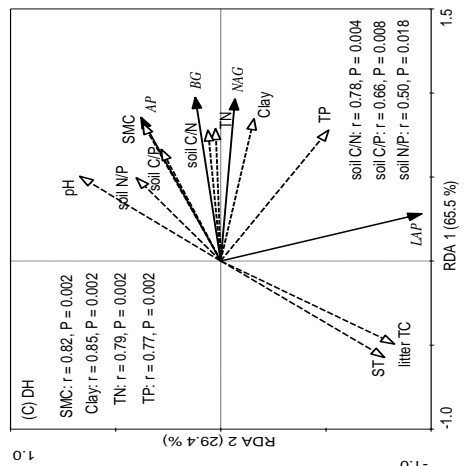

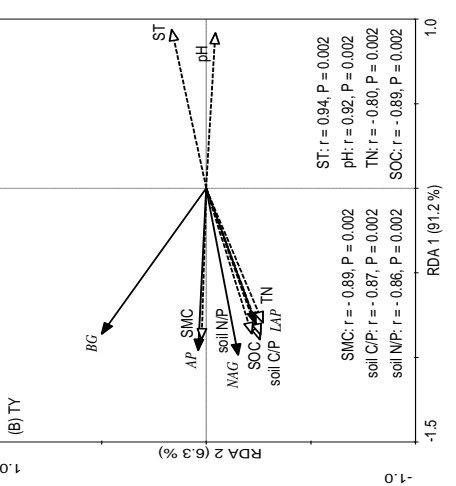

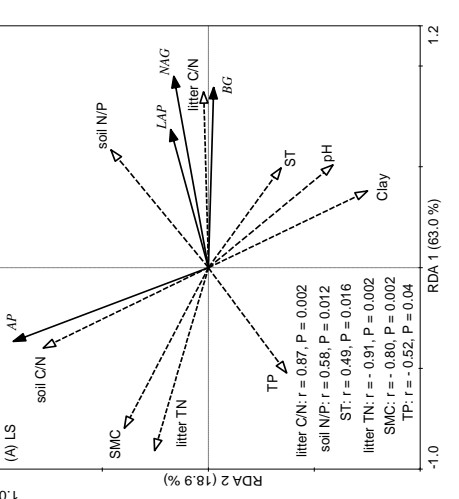


**Figure 4.** Redundancy analysis (RDA) ordination biplot of soil enzyme activities and environmental properties for the different forest types in different climatic zones (A. Liangshui; B. Taiyue;
C. Dinghu). Only the environmental variables that were significantly correlated with RDA1 are shown. The dotted lines and solid lines represent the environmental variables and enzyme
activities. The variables in this table were abbreviated as follows: TC(litter) = litter total carbon; TN(litter) = litter total nitrogen; C/N(litter) = litter total carbon/nitrogen; ST = soil temperature;
SMC = soil moisture content; Clay = soil clay content; SOC = soil organic carbon; TN = soil total nitrogen; TP = soil total phosphorus; C/N = soil carbon/nitrogen; C/P = soil carbon/phosphorus,
and N/P = soil nitrogen/phosphorus.





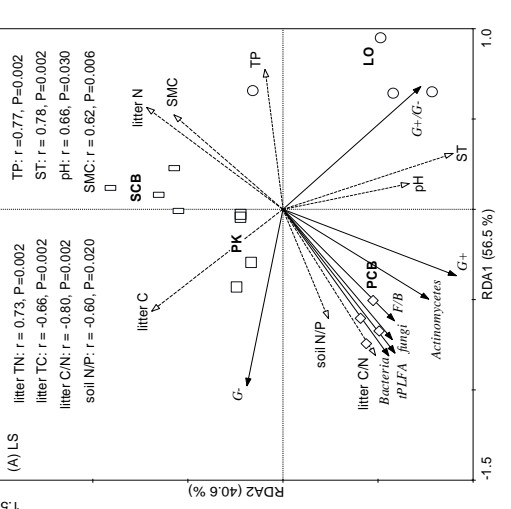

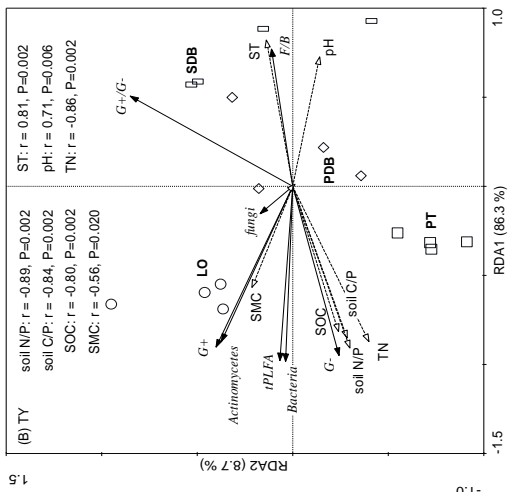

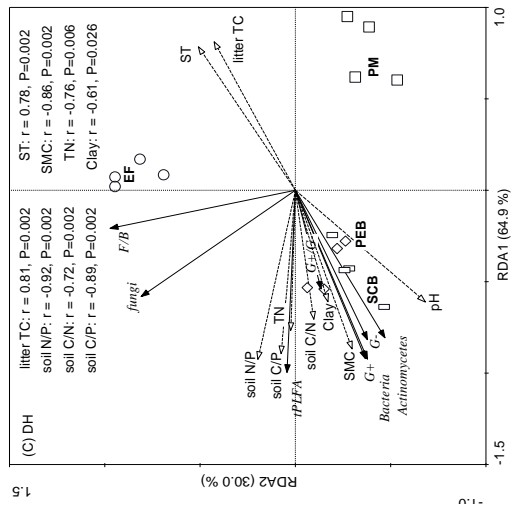


**Figure 5.** Redundancy analysis (RDA) ordination biplot of soil microbial community structure and environmental properties for different forest types in different climatic zones (A. Liangshui;
B. Taiyue; C. Dinghu). Only the environmental variables that were significantly correlated with RDA1 are shown. The dotted lines and solid lines represent the environmental variables and lipid
signatures. The abbreviations of the variables included in this figure are shown in Figure 4.