# Peer review of "Divergence of dominant factors on soil microbial 1 communities and functions in forest ecosystems along a climatic 2 gradient 3 4 Zhiwei Xu1,2, Guirui Yu3,4,\*, Xinyu Zhang3,4,\*, Nianpeng He3,4, Qiufeng Wang3,4, Shengzhong 5 Wang 1,2"

_Biogeosciences, 2017_

## Referee Comment (RC1) · Anonymous Referee #1 · 28 Sep 2017

Divergence of dominant factors on soil microbial communities and functions in forest ecosystems along a climatic gradient is a investigation paper. Authors chose 12 forests along three climate zones to investigate the variation of soil activities and microbe structures among these forests along three climate zones. The results showed that soil enzyme activities and microbial PLFAs differed with forest types along climatic zones. Both climate and forest type had significant effects on soil enzyme activities and microbial communities. Litter nutrients made an important effect to variations in the soil microbial communities and enzyme activities in temperate zones, while soil micro-climate and nutrients were the main effect factors on the soil microbial community structure and enzymatic activities in warm temperate and subtropical zones. The

paper has valuable to be published in this journal. However, the following points should be considered to reviseïijŽ 1, Abstract: line 44-45 "Our results indicate that the main controls on soil microbes and functions vary across forest ecosystems in different climatic zones, and that the effects of soil moisture content, soil temperature, and the soil N/P ratio were considerable." was the results, not indications. Insteadly, please give a general summery about reasons of the variation. 2ïijŇMaterials and Method The investigation was conducted in July and August in three climate zones, It is better to illustrate the climate information of the investigation month and detail investigation date in each site. This is because the activities of microbe is very sensitive to the climates, especially the moisture and temperature. 3, Results In the 3.1 section, the activities of four enzymes did not be described carefully. Most informations were ignored, for example, there were not comparison between forest types in the same climate zone. And there were not comparison between different climate zones, for example,the LAP activities of microbes in worm temperate zone were much higher than that in temperate zone. In the 3.2, same as 3.1, no comparison among three climate zones. Although there were difference among forest types in the same zone, authors should compare the similar quality forest such as SCB along three climate zones. If the results of these comparison could be reported, more mechanism of divergences among zones and forest types could be understood very well. In conclusion part, we would like to see the conclusion what changes along the climate zone could be found 4, Discussion 4.1,It is unclear what the response of soil enzyme activies and microbial plfas to variation of forest types is. Authors should clearly discuss the variation partten and formation reseason. 4.2, How to compare the commen effect and key effect? if there is obviose differece between two effects, could you explain the identification method of two effects.. 5.Conclusion Authors should adress the main conclution of the variation of enzyme actiivities and microbial communnity among forest types along the three zones in the suitable part of the paragraph. Minor mistakes 1. line 135. authors should give detail information about collection such as which samples were collected in July? 2. SCB in temperate zone was not same as it in Subtropical forest, it is better abbreviated

as SCBt SCBs 3. Fig. 2 ABCD was represent different enzyme activities, please check them 4. The format of some references did not fit with the format of this journal such as New Physiologist which did not was abbreviated.

---

## Referee Comment (RC2) · Anonymous Referee #2 · 11 Oct 2017

The authors present a comprehensive study of soil microbial communities and extracellular enzyme activities in different forests along a climatic gradient. The methods are technically sound. This paper clearly elucidates the dominant factors controlling microbial communities and enzyme activities in each climatic zone. The authors also attempt to emphasize the importance of climatic zones in addition to forest types. However, it's unclear for readers why different dominant factors exhibit in different climatic zones. For example, the authors state that "soil clay content had most influence on the soil enzyme activities in subtropical forests" (Line 353). However, the following discussion is very general and does not explain why this is only found in the subtropics. Here is another example, soil nutrients (N, P) are more important in warm temperate and
subtropical forests than in temperate forests, because nutrients are more likely limiting factors in warm temperate and subtropical forest. This kind of comparison between different climatic zones should be expanded in Discussion and could add value to this study. I have a few more suggestions to improve the presentation of this study: In Conclusions, soil clay fraction is identified as an important predictor in subtropical zones. However, "soil clay" is not mentioned in Abstract. Line 266-268: I don't understand the logic here. The authors are talking about microbial/enzyme responses to forest types in Section 4.1. The concluding sentence addresses "climatic region may be more important than forest types" without any expanded discussion, though I understand "climatic effects" may be indirectly discussed in Section 4.3. Line 298-300: This clause does not explain why there are more Gram-negative bacteria, less Gram-positive bacteria, and (less?) bacteria PLFAs under increasing pH.

Minor comments: Line 210-212: please spell out G- (Gram-negative bacteria) and G+ (Gram-positive bacteria) when they are first introduced. Line 241-243: The causal explanation herein is not specifically related to the results in Section 3.3 and Fig. 4a. Does the "higher inputs of mixed litter" mean higher litter C/N and lower litter TN? To my understanding, from Fig.4a, BG/NGC/LAP activities are positively correlated with litter C/N and negatively correlated with litter TN. The following explanation for the warm temperate zone is more informative. Line 262: please spell out SLA and LDMC. Line 328: please spell out F/B ratio.

---

## Author Comment (AC1) · 12 Nov 2017

Reviewer #1: Interactive comment on "Divergence of dominant factors on soil microbial communities and functions in forest ecosystems along a climatic gradient" by Zhiwei Xu et al. Anonymous Referee #1 Divergence of dominant factors on soil microbial communities and functions in forest ecosystems along a climatic gradient is a investigation paper. Authors chose 12 forests along three climate zones to investigate the variation of soil activities and microbe structures among these forests along three climate zones. The results showed that soil enzyme activities and microbial PLFAs differed with forest types along climatic zones. Both

climate and forest type had significant effects on soil enzyme activities and microbial communities. Litter nutrients made an important effect to variations in the soil microbial communities and enzyme activities in temperate zones, while soil micro-climate and nutrients were the main effect factors on the soil microbial community structure and enzymatic activities in warm temperate and subtropical zones. The C1 BGD Interactive comment Printer-friendly version Discussion paper has valuable to be published in this journal. However, the following points should be considered to revise.

(1) Abstract: line 44-45 "Our results indicate that the main controls on soil microbes and functions vary across forest ecosystems in different climatic zones, and that the effects of soil moisture content, soil temperature, and the soil N/P ratio were considerable." was the results, not indications. Instead, please give a general summery about reasons of the variation.

AN: we have improved this part as "Our results showed that the main controls on soil microbes and functions vary in different climatic zones, and that the effects of soil moisture content, soil temperature, clay content, and the soil N/P ratio were considerable." (P2, Line 46-49).

(2) Materials and Method: The investigation was conducted in July and August in three climate zones, It is better to illustrate the climate information of the investigation month and detail investigation date in each site. This is because the activities of microbe is very sensitive to the climates, especially the moisture and temperature.

AN: We illustrated the climate information of the investigation month in the text. The average temperature of the sampling month was 21.3 °C, 17.4°C, 27.3°C with the relative humidity of 78%, 60-65%, 83.5% in LS, TY, and DH, respectively. The sampling dates are Jul.5 2013, Jul.28 2013, Aug.15 2013 in LS, TY, and DH, respectively. (P5, Line133-136).

(3) Results: In the 3.1 section, the activities of four enzymes did not be described carefully. Most information were ignored, for example, there were not comparison between

forest types in the same climate zone. And there were not comparison between different climate zones, for example, the LAP activities of microbes in worm temperate zone were much higher than that in temperate zone.

AN: We have added necessary description about the enzyme activities. The soil BG and NAG activities were much higher in the coniferous forest than in the conifer broad-leaved mixed forests and the broad-leaved forests (Table S2). The soil AP enzyme activities were highest in the conifer broad-leaved mixed forests and lowest in the coniferous forests (Table S2). (P8, Line205-208).

The soil BG, NAG, and LAP activities were much higher in the warm temperate zone than in the temperate and the subtropical climate zones (Table S2). The AP activities were highest in the subtropical climate zone (Table S2). (P8, Line 210-212)

(4) In the 3.2, same as 3.1, no comparison among three climate zones. Although there were difference among forest types in the same zone, authors should compare the similar quality forest such as SCB along three climate zones. If the results of these comparison could be reported, more mechanism of divergences among zones and forest types could be understood very well.

AN: We have added necessary description about the microbial communities. We compare the microbial PLFAs among the three different climate zones and three forest types (conifer broad-leaved mixed forest, broad-leaved forest, and coniferous forest), respectively.

The forest type had a significant effect on the soil bacteria, fungi, gram-positive bacteria (G+), and gram-negative bacteria (G−) PLFAs (Table 2). The soil total PLFAs, bacteria, G+, G−, and actinomycete were much higher in the conifer broad-leaved mixed forests than in the coniferous forests and the broad-leaved forests (Table S2). The soil fungi was highest in the broad-leaved forest and lowest in the coniferous forest (Table S2). (P8, Line221-226).

With the exception of the soil G+/ G−, the effects of the combination of climate and forest type on all soil PLFAs were significant, and were stronger than the individual effects of either climate or forest type (Table 2, Table S2). Climate had a significant effect on the total PLFAs, fungi, and G− (P<0.0001) (Table 2). The soil total PLFAs, bacteria, G+, and G− were much higher in the temperate zone than in the warm temperate and the subtropical zones (Table S2). The fungi, F/B, and G+/G− were highest in the subtropical zone (Table S2). (P9, Line 227-232)

(5) In conclusion part, we would like to see the conclusion what changes along the climate zone could be found

AN: We have added description in the result and conclusions part about the variations in microbial communities and enzyme activities along the climate zone.

The soil BG, NAG, and LAP activities were much higher in the warm temperate zone than in the temperate and the subtropical climate zones (Table S2). The AP activities were highest in the subtropical climate zone (Table S2). (P8, Line 210-212)

With the exception of the soil G+/ G−, the effects of the combination of climate and forest type on all soil PLFAs were significant, and were stronger than the individual effects of either climate or forest type (Table 2, Table S2). Climate had a significant effect on the total PLFAs, fungi, and G− (P<0.0001) (Table 2). The soil total PLFAs, bacteria, G+, and G− were much higher in the temperate zone than in the warm temperate and the subtropical zones (Table S2). The fungi, F/B, and G+/G− were highest in the subtropical zone (Table S2). (P9, Line 227-232)

Conclusion: Except AP, soil enzyme activities were highest in warm temperate zone. Soil tPLFAs, bacteria, G− increased from temperate zone to subtropical zone, but fungi was in reverse. (P15, Line 404-406).

(6) Discussion: 4.1, It is unclear what the response of soil enzyme activies and microbial plfas to variation of forest types is. Authors should clearly discuss the variation

partten and formation reseason.

AN: We have improved this part. Forests in the same climate zone developed similar microbe functions which confirmed the result that the effect of climate on soil enzyme activities were stronger than the forest type and their interactive effect. However, there were still differences among the enzyme activities in different forest types of the same climate zone. Soil microorganisms are usually considered to be C limited, and the litter inputs with high C/N ratio of PCB in the temperate zone will stimulate microbes to grow and secrete more enzymes (Table 1). Therefore, all enzyme activities were highest in PCB in the temperate zone. (P10, Line 264-270).

The high soil BG enzyme activities in the LOw forest in the warm temperate zone reflect the litter inputs with low C. Because that soil enzyme activities will not continuously increase or decrease as nutrient availability increases or decreases. When the soil nutrients are short in supply, microbes will potentially increase production of nutrient-acquiring enzymes, because they are expected to optimize the allocation of their resource reserves by acquiring the resource that is most limiting (Bloom et al., 1985). (P10, Line 270-275).

The interactive effect of climate and forest type were more important than the individual effect of them. Therefore, the soil microbial communities of the 12 forests were separated from each other. Vegetation transfers substrate material of varying quality to microbes through litter fall. Fungi are more suitable for life in environments containing higher C/N ratios and low soil pH (Nilsson et al., 2012). The four broadleaved forests were high in litter C/N ratio (Table 1). Therefore, fungi were dominated in this harsh nutrient environments and highest in broadleaved forests. The litter and soil from conifer broad-leaved mixed forest were high in C, N, and P, and promotes the propagation of bacteria that favor high-nutrient soil (Priha and Smolander, 1997; Priha et al., 2001). Therefore, the structures and functions of the soil microbial communities that developed in the different types of forest were unique. (P10, Line 277-283; P11, Line 284-286)

To avoid the repetition with the 4.2 and 4.3, some more detail reasons of the variations were discussed later.

(7) 4.2, How to compare the commen effect and key effect? if there is obviose differece between two effects, could you explain the identification method of two effects.

AN: The common effect refer to the same environmental variables which are significantly correlated with the RDA1 in the three bioplots of the three climate zones (P<0.05). The key effect refer to the environmental variables those were more important in determining soil microbial communities and functions of the individual climate zones (P<0.01).

In addition, we have done a new RDA again by putting the data of 12 forests in the three climate zones together to observe the variations in soil enzyme activities (Fig.S1) and microbial communities (Fig.S2) among different forest types and climate zones.

(8) Conclusion: Authors should adress the main conclution of the variation of enzyme actiivities and microbial communnity among forest types along the three zones in the suitable part of the paragraph.

AN: We have added the main conclusion of the variations of enzyme activities and microbial community among forest types in the result and conclusion.

The soil total PLFAs, bacteria, G+, G−, and actinomycete were much higher in the conifer broad-leaved mixed forests than in the coniferous forests and the broad-leaved forests. The soil BG and NAG activities were much higher in the coniferous forest than in the conifer broad-leaved mixed forests and the broad-leaved forests. Except AP, soil enzyme activities were highest in warm temperate zone. Soil tPLFAs, bacteria, G− increased from temperate zone to subtropical zone, but fungi was in reverse. (P15, Line 401-406)

Minor mistakes

(9) line 135. authors should give detail information about collection such as which

samples were collected in July?

AN: The average temperature of the sampling month was 21.3 °C, 17.4°C, 27.3°C with the relative humidity of 78%, 60-65%, 83.5% in LS, TY, and DH, respectively. The sampling dates are Jul.5 2013, Jul.28 2013, Aug.15 2013 in LS, TY, and DH, respectively. (P5, line 133-136; Table 1).

(10) SCB in temperate zone was not same as it in Subtropical forest, it is better abbreviated as SCBt SCBs

AN: DONE (Table 1, Figure 1, Figure 2 and Figure 4).

(11) Fig. 2 ABCD was represent different enzyme activities, please check them.

AN: DONE (P24, Figure 1).

(12) The format of some references did not fit with the format of this journal such as New Physiologist which did not was abbreviated.

AN: DONE (P16, Line 436). We have checked all through the text and made necessary variations.

Please also note the supplement to this comment:
https://www.biogeosciences-discuss.net/bg-2017-243/bg-2017-243-AC1-supplement.zip
* * *
[Figure]

**Figure 1.** Soil enzyme activities under different forest types in different climatic zones. BG, b-1, 4-glucosidase;

NAG, b-1,4-N-acetylglucosaminidase; LAP, leucine aminopeptidase; AP, acid phosphatase. The capital letters A,

B, C, and D represent the variations in the enzyme activities of BG, NAG, LAP and AP, respectively. Different

lowercase letters indicate significant differences between forests in the same climatic zone. The abbreviations of

the sampling sites are shown in Table 1.

**Fig. 1.** Soil enzyme activities under different forest types in different climatic zones.

[Figure]

**Figure 2.** The PLFA contents, Fungi:Bacteria ratios, and G⁺/G⁻ for different forest types in different climatic zones

(A. Liangshui; B. Taiyue; C. Dinghu). Different lowercase letters indicate significant differences among forests in

the same climatic zone. F/B, fungi/bacteria; G⁺/G⁻, Gram-positive bacteria/ Gram-negative bacteria. The

abbreviations of the sampling sites are shown in Table 1.

**Fig. 2.** The PLFA contents, Fungi:Bacteria ratios, and G+/G− for different forest types in different climatic zones (A. Liangshui; B. Taiyue; C. Dinghu).

[Figure]

**Figure 3.** Redundancy analysis (RDA) ordination biplot of soil enzyme activities and environmental properties for the different forest types in different climatic zones (A. Liangshui; B. Taiyue; C. Dinghu). Only the environmental variables that were significantly correlated with RDA1 are shown. The dotted lines and solid lines represent the environmental variables and enzyme activities. The variables in this table were abbreviated as follows: TC(litter) = litter total carbon; TN(litter) = litter total nitrogen; C/N(litter) = litter total carbon/nitrogen; ST = soil temperature; SMC = soil moisture content; Clay = soil clay content; SOC = soil organic carbon; TN = soil total nitrogen; TP = soil total phosphorus; C/N = soil carbon/nitrogen; C/P = soil carbon/phosphorus, and N/P = soil nitrogen/phosphorus.

**Fig. 3.** Redundancy analysis (RDA) ordination biplot of soil enzyme activities and environmental properties for the different forest types in different climatic zones (A. Liangshui; B. Taiyue; C. Dinghu).

[Figure]

**Figure 4.** Redundancy analysis (RDA) ordination biplot of soil microbial community structure and environmental properties for different forest types in different climatic zones (A. Liangshui;

B. Taiyue; C. Dinghu). Only the environmental variables that were significantly correlated with RDA1 are shown. The dotted lines and solid lines represent the environmental variables and lipid

signatures. The abbreviations of the variables included in this figure are shown in Figure 4.

**Fig. 4.** Redundancy analysis (RDA) ordination biplot of soil microbial community structure and environmental properties for different forest types in different climatic zones (A. Liangshui; B. Taiyue; C. Ding

---

## Author Comment (AC2) · 12 Nov 2017

Reviewer 2

(1) The authors present a comprehensive study of soil microbial communities and extracellular enzyme activities in different forests along a climatic gradient. The methods are technically sound. This paper clearly elucidates the dominant factors controlling microbial communities and enzyme activities in each climatic zone. The authors also attempt to emphasize the importance of climatic zones in addition to forest types. However, it's unclear for readers why different dominant factors exhibit in different climatic zones. For example, the authors state that "soil clay content had most influence on the

soil enzyme activities in subtropical forests" (Line 353). However, the following discussion is very general and does not explain why this is only found in the subtropics.

AN: We have improved this part. Therefore, soil enzyme activities and microbial PLFAs were highest in the SCBs forest with finely texture. Except SCBt in the temperate zone and PT in the warm temperate zone, the soil clay content were not significant different among other three forest types. However, the soil clay contents of the four forest types in the subtropical zone were significant different from each other and important for variations in microbial communities and functions (Table 1). (P14, Line 376-381).

(2) Here is another example, soil nutrients (N, P) are more important in warm temperate and subtropical forests than in temperate forests, because nutrients are more likely limiting factors in warm temperate and subtropical forest. This kind of comparison between different climatic zones should be expanded in Discussion and could add value to this study.

AN: We have improved this part as "The soil TN and TP were lower in the warm temperate and subtropical zone than in the temperate zone in our study (Table 1), and these two kinds of nutrients were more likely limiting factors in warm temperate and subtropical forest (DeForest et al., 2012; Xu et al., 2017). Therefore, soil TN and TP are more important in warm temperate and subtropical forests than in temperate forests." (P13, Line 354-357).

(3) I have a few more suggestions to improve the presentation of this study: In Conclusions, soil clay fraction is identified as an important predictor in subtropical zones. However, "soil clay" is not mentioned in Abstract.

AN: We have improved the abstract. Our results showed that the main controls on soil microbes and functions vary in different climatic zones, and that the effects of soil moisture content, soil temperature, clay content, and the soil N/P ratio were considerable. (P2, Line 47-50).

(4) Line 266-268: I don't understand the logic here. The authors are talking about microbial/enzyme responses to forest types in Section 4.1. The concluding sentence addresses "climatic region may be more important than forest types" without any expanded discussion, though I understand "climatic effects" may be indirectly discussed in Section 4.3.

AN: We have moved this sentence to the section 4.2 and improved it as "This was also demonstrated by the stronger effect of climate on soil enzyme activities and the combined interaction effect of climate and forest type on soil microbial communities. Other studies have reported that precipitation and mean annual temperature played important roles in explaining on the large-scale distribution of soil microbial community composition and functions (de Vries et al., 2012; Xu et al., 2017)." (P11, Line 311-312; P12, Line 313-316).

(5) Line 298-300: This clause does not explain why there are more Gram-negative bacteria, less Gram-positive bacteria, and (less?) bacteria PLFAs under increasing pH.

AN: We have improved this part as "Soil G+/G− ratios were highest in the subtropical forest where G− bacteria PLFAs were least abundant, which may reflect microbial growth strategies. The G+ bacteria are primarily K-strategists that can survive over long periods in the soil under harsh conditions with lower soil pH (Andrews & Hall, 1986). Increased pH causes an increase in bacterial diversity and a shift in the bacterial community to more G− and fewer G+ bacteria PLFAs (Wu et al., 2009; Shen et al., 2013). "(P12, Line 321-326).

(6) Line 210-212: please spell out G- (Gram-negative bacteria) and G+ (Gram-positive bacteria) when they are first introduced.

AN: DONE (P8, Line 222-223).

(7) Line 241-243: The causal explanation herein is not specifically related to the results

in Section 3.3 and Fig. 4a. Does the "higher inputs of mixed litter" mean higher litter C/N and lower litter TN? To my understanding, from Fig.4a, BG/NGC/LAP activities are positively correlated with litter C/N and negatively correlated with litter TN. The following explanation for the warm temperate zone is more informative.

AN: We have improved this part as "Soil microorganisms are usually considered to be C limited, and the litter inputs with high C/N ratio of PCB in the temperate zone will stimulate microbes to grow and secrete more enzymes (Table 1). Therefore, all enzyme activities were highest in PCB in the temperate zone." (P10, Line 267-270).

(8) Line 262: please spell out SLA and LDMC.

AN: We have deleted this part.

(9) Line 328: please spell out F/B ratio.

AN: DONE (P13, Line 342).

Please also note the supplement to this comment:
https://www.biogeosciences-discuss.net/bg-2017-243/bg-2017-243-AC2-
supplement.zip
* * *
[Figure]

[Figure]

**Figure 1.** Soil enzyme activities under different forest types in different climatic zones. BG, b-1, 4-glucosidase;

NAG, b-1,4-N-acetylglucosaminidase; LAP, leucine aminopeptidase; AP, acid phosphatase. The capital letters A,

B, C, and D represent the variations in the enzyme activities of BG, NAG, LAP and AP, respectively. Different

lowercase letters indicate significant differences between forests in the same climatic zone. The abbreviations of

the sampling sites are shown in Table 1.

**Fig. 1.**

[Figure]

**Figure 2.** The PLFA contents, Fungi:Bacteria ratios, and G+/G− for different forest types in different climatic zones

(A. Liangshui; B. Taiyue; C. Dinghu). Different lowercase letters indicate significant differences among forests in

the same climatic zone. F/B, fungi/bacteria; G+/G−, Gram-positive bacteria/ Gram-negative bacteria. The

abbreviations of the sampling sites are shown in Table 1.

**Fig. 2.**

[Figure]

**Figure 3.** Redundancy analysis (RDA) ordination biplot of soil enzyme activities and environmental properties for the different forest types in different climatic zones (A. Liangshui; B. Taiyue; C. Dinghu). Only the environmental variables that were significantly correlated with RDA1 are shown. The dotted lines and solid lines represent the environmental variables and enzyme activities.

The variables in this table were abbreviated as follows: TC(litter) = litter total carbon; TN(litter) = litter total nitrogen; C/N(litter) = litter total carbon/nitrogen; ST = soil temperature; SMC = soil moisture content; Clay = soil clay content; SOC = soil organic carbon; TN = soil total nitrogen; TP = soil total phosphorus; C/N = soil carbon/nitrogen; C/P = soil carbon/phosphorus, and N/P = soil nitrogen/phosphorus.

**Fig. 3.**

[Figure]

**Figure 4.** Redundancy analysis (RDA) ordination biplot of soil microbial community structure and environmental properties for different forest types in different climatic zones (A. Liangshui; B. Taiyue; C. Dinghu). Only the environmental variables that were significantly correlated with RDA1 are shown. The dotted lines and solid lines represent the environmental variables and lipid signatures. The abbreviations of the variables included in this figure are shown in Figure 4.

**Fig. 4.**